# Rapid Green Synthesis and Characterization of Silver Nanoparticles Arbitrated by Curcumin in an Alkaline Medium

**DOI:** 10.3390/molecules24040719

**Published:** 2019-02-16

**Authors:** Muhammad Jamshed Khan, Kamyar Shameli, Awis Qurni Sazili, Jinap Selamat, Suriya Kumari

**Affiliations:** 1Institute of Tropical Agriculture and Food Security, Universiti Putra Malaysia (UPM), Serdang 43400, Malaysia; gs50508@student.upm.edu.my (M.J.K.); awis@upm.edu.my (A.Q.S.); jinap@upm.edu.my (J.S.); 2Faculty of Veterinary Sciences, Bahauddin Zakariya University, Multan 60800, Pakistan; 3Malaysia-Japan International Institute of Technology, Universiti Teknologi Malaysia, Jalan Sultan Yahya Petra, Kuala Lumpur 54100, Malaysia; kamyarshameli@gmail.com

**Keywords:** alkaline medium, curcumin silver nanoparticles, green synthesis, meat preservation

## Abstract

Green synthesis of silver nanoparticles is desirable practice. It is not only the required technique for industrial and biomedical purposes but also a promising research area. The aim of this study was to synthesize green curcumin silver nanoparticles (C-Ag NPs). The synthesis of C-Ag NPs was achieved by reduction of the silver nitrate (AgNO_3_) in an alkaline medium. The characterizations of the prepared samples were conducted by ultraviolet visible (UV-vis) spectroscopy, powder X-ray diffraction (PXRD), field emission scanning electron microscopy (FESEM), high-resolution transmission electron microscopy (HRTEM), selected area electron diffraction (SAED) and zeta potential (ZP) analyses. The formation of C-Ag NPs was evaluated by the dark color of the colloidal solutions and UV-vis spectra, with 445 nm as the maximum. The size of the crystalline nanoparticles, recorded as 12.6 ± 3.8nm, was confirmed by HRTEM, while the face-centered cubic (fcc) crystallographic structure was confirmed by PXRD and SAED. It is assumed that green synthesized curcumin silver nanoparticles (C-Ag NPs) can be efficiently utilized as a strong antimicrobial substance for food and meat preservation due to their homogeneous nature and small size.

## 1. Introduction

Nanotechnology is the utilization of nanoparticles for biological, medical, and processing purposes. Nanoparticles can be naturally found in the form of ash, biomolecules, soil particles or artificially engineered for the usage in various enterprises [1]. Green synthesized metal nanoparticles have been studied by researchers due to their unmatched characteristics [2]. As a result, the utilization of nanoparticles has been increased in food packaging, coatings, pharmaceutical utilization, and biological tagging [3,4].

Silver nanoparticles (Ag NPs), have gained the attention of researchers and scientists due to their wide range of applications as antimicrobial, antifungal, antioxidant, anticancer, and anti-inflammatory agents [5,6,7,8]. Ag NPs can tolerate a high temperature range with very low volatility as compared to other nanoparticles [9], and can restrain the growth of micro organisms following initial contact with them [10]. Currently, the concept of “active packaging” is well known for extending the shelf life of food. Active packaging includes the incorporation of nanoparticles (Ag, Zn, Mg, etc.) into edible films to inhibit the growth of targeted microbes for long -term preservation [11]. The toxicity of Ag NPs is mainly related to their size and even distribution when incorporated into edible films [11]. Ag NPs with a size of less than 20 nm can exhibit more antimicrobial ability by penetrating into the microbial membranes causing the inhibition of ATP (Adenosine triphosphate) and DNA (Deoxyribonucleic acid) replication [11]. The synthesis of Ag NPs has been practiced using various physical and chemical routes, but their non-toxic, economical, and benign biological characteristics have captured the attention of the researchers [12,13,14,15,16]. The biological and chemical compounds, used for Ag NPs synthesis, include polysaccharides, microbes, microbial products, actinomycetes, and various plant extracts [17,18]. A new and relatively safer method has been provided by Raveendran et al. [19] to produce Ag NPs by “green synthesis”.

Curcumin is one of the most useful plant-based materials derived from “turmeric”, having the potential for the green synthesis of Ag NPs. This is due to “polyphenol”, which may trigger the synthesis of Ag NPs during the reduction process [20]. However, the application of curcumin has been limited in the production of Ag NPs due to low water solubility and sensitivity to heat, light, and alkaline pH [21]. Various methods have been reported for the green synthesis of Ag NPs from curcumin. These methods include ultrasonication, chemical reduction, physical method, physio-chemical synthesis, and chemical mediator (dichloromethane, DMSO, Na_2_CO_3_) technique [20,21,22,23,24,25,26,27]. Time consumption [20], toxicity of the chemicals used as mediator [22,24,27] low solubility of curcumin [21,26], low production rate [20,24], cost consumption, and large size of Ag NPs [22,23] were the major drawbacks of these methods. The eco-friendly [28], facile, cost effective, and green [29,30,31,32] methods have also been reported for the synthesis of Ag NPs from curcumin. In this paper, we reported the rapid green synthesis of curcumin silver nanoparticles (C-Ag NPs) in an alkaline medium which can be considered green, eco-friendly and safe for producing nanoparticles as an antibacterial substance for food processing. Silver nitrate was used as the silver precursor, while curcumin was used as the reducing and capping agent. The impact of curcumin weight on the crystalline size of the synthesized Ag NPs was also examined.

## 2. Results and Discussion

The reduction of curcumin dissolved in an alkaline solution of sodium hydroxide (NaOH; 1M) was triggered by mixing curcumin oxide solution into an aqueous solution of silver nitrate (AgNO_3_). The reduction rate was directly proportionate to the increase in weight of the curcumin in NaOH (1M) solution i.e., from 0.05 g to 0.25 g.

The reduction rate of curcumin was prominent in a 0.25 M AgNO_3_ solution, with a characteristic black color of the colloidal solution. The following equations can demonstrate this reduction process:(1)$${\mathrm{Ag}}_{\left(\mathrm{aq}\right)}^{+}\;+\;\mathrm{curcumin}\overset{\mathrm{Stirring}\;\mathrm{at}\;25\;{}^{\circ}C}{\to }{\left[\mathrm{Ag}\left(\mathrm{curcumin}\right)\right]}^{+}$$
(2)$${\left[\mathrm{Ag}\left(\mathrm{curcumin}\right)\right]}^{+}\;+\;\text{R-CHO}\overset{\mathrm{Stirring}\;\mathrm{at}\;25\;{}^{\circ}C}{\to }{\left[\mathrm{Ag}\left(\mathrm{curcumin}\right)\right]}^{+}\;\text{R-COOH}$$

The silver nitrate produced dispersed Ag+ ions in aqueous form, which combined with curcumin to form [Ag(curcumin)]^+^ complex (Equation (1)). The [Ag(curcumin)]^+^ complex further reacted with an aldehyde group (from methanolic group of curcumin chemical structure) to form C-Ag through reduction of Ag^+^ ions (Figure 1) [26]. The reduction of Ag^+^ ions was due to the oxidation of the aldehyde group into carboxylic acid group (Equation (2)) [26].

### 2.1. UV-Visible Spectroscopy of C-Ag NPs

The colloidal solutions showed the color changes after the addition of 0.75 mL curcumin oxide solution (C0, C1, C2) into 100 mL aqueous solutions of AgNO_3_ (0.25 M). The changes in the color of colloidal solutions confirmed the presence of C-Ag NPs [31]. The optimal results were recorded when curcumin oxide (C) solutions (C0, C1, C2) were added into the concentration of 0.25 M compared to 0.125 M and 0.50 M concentrations of AgNO_3_.

The color of the solution turned darker as the weight of curcumin increased in the NaOH solution, i.e., 0.05 g to 0.25 g. The color changes are recorded in Figure 2.

The changes in color indicated the excitation of the surface plasmon vibration in the metal nanoparticles [33] formed during green synthesis. The changes in the color of the obtained colloidal solution confirmed the formation of C-Ag NPs, which was later confirmed by UV-visible spectroscopy. The UV-visible spectra were recorded for all three samples of curcumin oxide (C0, C1 and C2). Curcumin-reduced Ag NPs showed absorbance spectra at 445 nm for C2, 428 nm for C1, and 427 nm for C0 samples (Figure 3). The sharp peak of absorbance at 445 nm for C2 samples depicts the formation of Ag NPs with almost spherical shapes, which was confirmed by high resolution transmission electron microscopy (HRTEM) and field emission scanning electron microscopy (FESEM). The results are in agreement with those reported by Abdelghany et al. [32], Pandit et al. [27] and Song et al. [21], where the nanoparticles exhibited a UV-vis spectral range of 220 nm to 800 nm.

For the current experiment, C-Ag NPs were synthesized at room temperature and these nanoparticles showed tremendous stability, for 180 days, in the solution.

### 2.2. Polydispersity Index (pdI) of Curcumin Samples

The polydispersity index (pdI) for C0 was 0.686, indicating this sample was more polydisperse for the analysis with poor quality. C1 and C2 had a polydispersity index (pdI) of 0.431 and 0.462, respectively, indicating the good quality of the solutions (Figure 4). The colloidal solutions having a pdI less than 0.50 are considered “good” quality [33]. These results were in agreement with the results reported by Huang [34], Basavaraja [4], and Mukherjee [35].

### 2.3. Structural Analysis of C-Ag NPs by Powder X-ray Diffraction (PXRD)

The structural analysis of Ag NPs obtained from C0, C1, and C2 samples, was confirmed by powder X-ray diffraction (PXRD) (Figure 5). The X-ray diffraction patterns of C0 and C1 samples showed peaks of AgNO_3_ along with the peaks of nano silver. These peaks might be due to the low reduction rate of AgNO_3_ as the concentrations of curcumin were low in C0 (0.05 g) and C1 (0.1 g) for the chemical reaction [20,25]. The X-ray diffraction pattern for the C-Ag NPs powder from C2 sample indicated a four peak pattern at 2 theta/Degree, i.e., 38.05°, 44.60°, 64.60° and 77.60° angles (Figure 5). The data reported for PXRD reference No. 00-004-0783 were properly matched with the crystalline nano silver (fcc, 99.9% pure) peaks, i.e., (111), (200), (220), and (311) produced by C-Ag NPs from C2 sample [29,30]. The crystallographic structure of C-Ag NPs, confirmed by PXRD, was “Face-Centered Cubic” (fcc).

The main crystalline phase in C-Ag NPs was silver and free from other impure phases in C2 sample. The average size of C-Ag NPs can be estimated by the Debye-Scherrer equation [35,36]
*n* = *Kλ*/*β·cos θ*(3)
where *n* indicates the size of the crystalline particles, K is the Sherrer constant (shape factor) with a value of 0.9to 1.0, *λ* is the X-ray wavelength (1.5424 for the powder sample), *β* is the line broadening at half of the XRD peak, and *θ* is the Bragg angle. By calculating *n* for C2 sample through the Debye-Scherrer equation the average size of crystalline C-Ag NPs was12.6 nm. This size is similar to the size confirmed by high resolution transmission electron microscopy (HRTEM).

The PXRD results are in agreement with reports by Verma et al. [20], Shameli et al. [26], Adibzadeh and Motakef-kazemi [29] and Al-Namil and Patra [30], of the crystalline phase, size, and structural analysis.

### 2.4. Field Emission Scanning Electron Microscopy (FESEM) of C-Ag NPs

Field emission scanning electron microscopy (FESEM) of crystalline C-Ag NPs (obtained from the C2 sample) was done to analyze the surface and shape. A dried powder of C-Ag NPs was used on the carbon-coated mounting grid. Three scanning scales were used to check the topography and the shape of the crystalline C-Ag NPs, i.e., 100,000×, 150,000×, and 200,000×. The clusters of C-Ag NPs were formed during the scanning analysis and spherical NPs were observed in SEM (Figure 6). The particle size, morphology, lattice surface analysis, and mean diameter of crystalline C-Ag NPs continued in HRTEM, as discussed later.

### 2.5. High-Resolution Transmission Electron Microscopy (HRTEM)

#### 2.5.1. Particle Size of Crystalline C-Ag NPs

The size and morphology of C-Ag NPs (obtained from C2 sample) were confirmed by high-resolution transmission electron microscopy (HRTEM). The morphology of C-Ag NPs was spherical (Figure 7). The mean diameter of crystalline C-Ag NPs was 12.6 ± 3.8 nm which was close to the measurement by PXRD analysis. All the nanoparticles were “homogeneous” surrounded by a faint and thin layer of organic capping material which might be curcumin [37,38]. The results were in agreement with those reported by El-Refai et al. [28], Alsammarraie et al. [29] and Kurian et al. [38], for the crystalline size less than 50 nm surrounded by the capping agent.

#### 2.5.2. Selected Area Electron Diffraction (SAED) and Surface Lattice Assessment of C-Ag NPs

Selected area electron diffraction (SAED) with different magnification confirmed the spherical crystalline nature of C-Ag NPs [38]. Figure 8 provides an over-view of the SAED pattern of C-Ag NPs. The peaks shown by PXRD results by C-Ag NPs can be observed from the core i.e., 111, 200, 220, 311 in SAED analysis where most of the spherical crystalline Ag NPs are present. The SAED results were in good agreement with PXRD results.

The lattice assessment of C-Ag NPs from C2 samples was done by HRTEM, confirming the “Quasi-spherical characteristics” of NPs [39]. The distance between the lattice fringes spacing of C-Ag NPs was recorded as 0.2663 nm from top to top (Figure 9). Furthermore, the lattice fringe spacing confirmed the face-centered cubic (fcc) nature of the C-Ag NPs [39], as discussed earlier in PXRD results. The results are in agreement with the report by Quester et al. [40].

#### 2.5.3. Zeta Potential (ZP) Analysis of C-Ag NPs

All the colloidal solutions were recorded as stable due to the zeta potential (ZP). The ZP was recorded as −47.5 mV (C0), −46.5 mV (C1), and −39.3 mV (C2) (Figure 10a,b), showing the high stability of the nano-suspension [33]. The negative ZP was due to the accumulation of negative charge at the surface and positive charge at the core. A minimum ±30 mV value of ZP is considered for the stability of a nano suspension, which is recorded by measuring the velocity of the nanoparticles in a DC electric field [26,27]. The mobility of the nanoparticles (µmcm/Vs) was −3.7 for C0, −3.6 for C1, and −3.0 for C2. The results are supported by the findings of Saha et al. [37] and Pandit et al. [27] regarding the negative ZP of the nanoparticles.

The summary of various characteristics of Ag NPs (including pdI, ZP, mobility, wavelength, and pH) synthesized from C2, C1, and C0 samples is given in (Table 1). It can be observed that the synthesis of C-Ag NPs, from C2 sample, was done at 25 °C with a pH of 9.92.

## 3. Materials and Methods

### 3.1. Materials

The materials for the synthesis of C-Ag NPs were curcumin (99.8% pure, anhydrous; Sigma-Aldrich, St. Louis, MO, USA), silver nitrate (AgNO_3_) (R & M, London, UK), deionized water (Sigma-Aldrich, St. Louis, MO, USA), and sodium hydroxide (NaOH; pellets; R & M, London, UK). All the reagents used in this study were of analytical grades and were used without further purification. The glass-ware was washed and cleaned thoroughly and autoclaved before use.

### 3.2. Green Synthesis of C-Ag NPs

Three concentrations of curcumin (C0, C1, and C2) were prepared following the procedure was adopted from Dhanaya, NP [41], with some modifications. In brief, curcumin concentrations were prepared by dissolving 0.05 g (C0), 0.1 g (C1), and 0.25 g (C2) pure curcumin (99.8%) into NaOH (1M) solution to form curcumin oxide solutions. Three concentrations of aqueous solution AgNO_3_ (0.125 M, 0.25 M, and 0.50 M) were prepared by dissolving the required mass of AgNO_3_ in deionized water.

In the synthesis of C-Ag NPs, 0.75 mL of curcumin oxide solution was added drop-wise into 100 mL aqueous solution of AgNO_3_ in a flask to formulate silver ions with curcumin. The color of the AgNO_3_ solutions changed immediately from clear to light brown then black, showing the formation of Ag NPs. The mixed solutions were stirred thoroughly for 2 h at 25 °C at 500 rpm to complete the chemical reaction. After stirring, the obtained solutions were centrifuged at 3800 rpm for 25 min at 28 °C to obtain nanoparticle pellets. The pellets were then washed thrice with deionized water to remove extra Ag^+^ ions and dried overnight at 35 °C for further study.

### 3.3. Characterization of C-Ag NPs

The characterization of C-Ag NPs was practiced through ultraviolet-visible spectroscopy (UV-vis) by using a UV-2600, UV-VIS spectrophotometer (SHIMADZU Corp., Kyoto, Japan) with a wavelength range of 220 nm to 800 nm. The structural analysis was done by powder X-ray diffraction (PXRD), (PANalytical EMPYREAN, Eindhoven, The Netherlands), Xpert High score Plus (V.14.0; PANalytical, Eindhoven, The Netherlands) at a scan speed of 2°/min with a lower angle of 30°–80°. The topographic images were taken by field emission scanning electron microscopy (FESEM) through JSM 7600 F FESEM with EXD (JEOL, Ltd., Tokyo, Japan) while high -resolution transmission electron microscopy (HRTEM) was done using a JEM 2100F filed emission electron microscope (JEOL, Ltd., Tokyo, Japan) with a resolution of 200 kV and excellent LaB_6_ electron gun. The zeta potential of C-Ag NPs was confirmed by a ZETASIZER Nano series (Nano ZS Malvern Instrument Ltd., Malvern, UK). The centrifugation of the colloidal solution was done by refrigerating micro centrifuge 5810 R (Sigma-Aldrich, Inc., St. Louis, MO, USA) to obtain C-Ag NPs pellets.

The colloidal solution dissolved in distilled water was used for most of the characterization procedures, while a C-Ag NPs pellets powder was used for PXRD and SEM.

### 3.4. Statistical Analysis

The data was statistically analyzed by SPSS software (V.20.0; IBM Corp., Armonk, NY, USA).

## 4. Conclusions

Curcumin silver nanoparticles were successfully synthesized by a green method in an alkaline medium with a pH of 9.92. The spectra maximum was recorded as 445 nm in UV-vis spectroscopy indicating the spherical shape of the nanoparticles. The spherical shape and crystalline size of NPs were confirmed by FESEM and HRTEM. The crystallographic structure of C-Ag NPs, confirmed by PXRD, was fcc, and they were free from any type of impurities with a crystalline size of 12.6 ± 3.8 nm. The zeta potential (ZP) was recorded as −39.3, proving the stability of nanoparticles in the colloidal solution after 180 days. Green synthesis of C-Ag NPs by a rapid, and very simple method is presented in this study, suggesting their multi-dimensional nano application in food processing, pharmaceutical, and medical enterprises. Furthermore, our study demonstrated that the size and the quality of C-Ag NPs could be optimized by varying the weight of curcumin, used as a reducing and capping agent, in the reduction process. However, the antibacterial and antimicrobial potential of these green synthesized C-Ag NPs is yet to be exploited in in-vitro studies as well as in edible polysaccharide films.

## Figures and Tables

**Figure 1 molecules-24-00719-f001:**
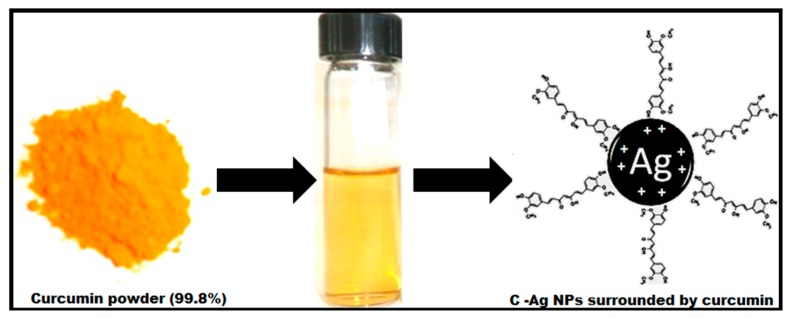
An illustration showing the formation of curcumin silver nanoparticles (C-Ag NPs) surrounded by curcumin.

**Figure 2 molecules-24-00719-f002:**
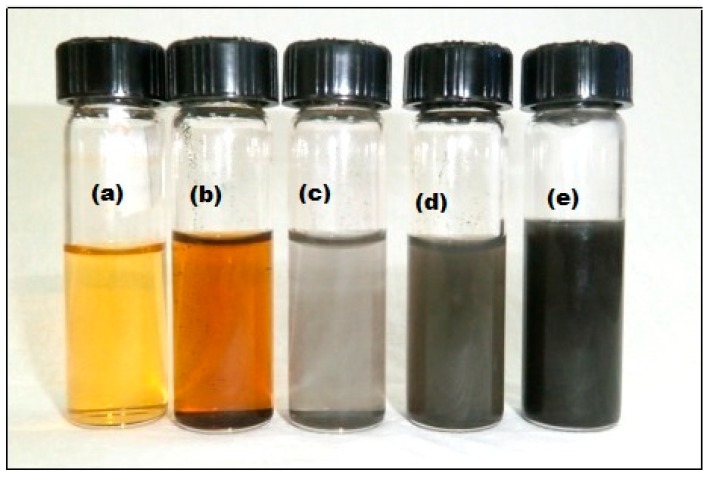
Changes in the color of the colloidal solution with various concentration of curcumin indicating the formation of C-Ag NPs: (**a**) curcumin solution (**b**) curcumin oxide (**c**) C0, 0.05 g curcumin (**d**) C1, 0.1 g curcumin (**e**) C2, 0.25 g curcumin.

**Figure 3 molecules-24-00719-f003:**
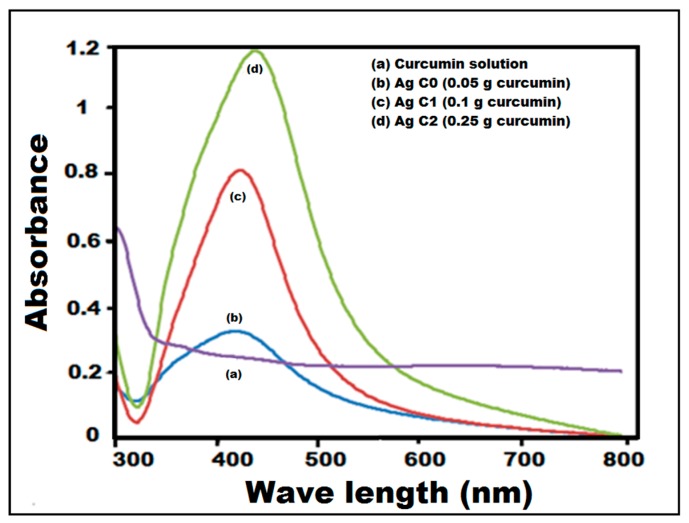
UV-visible absorption spectra for various concentrations of curcumin, i.e., Curcumin, C0, C1, and C2.

**Figure 4 molecules-24-00719-f004:**
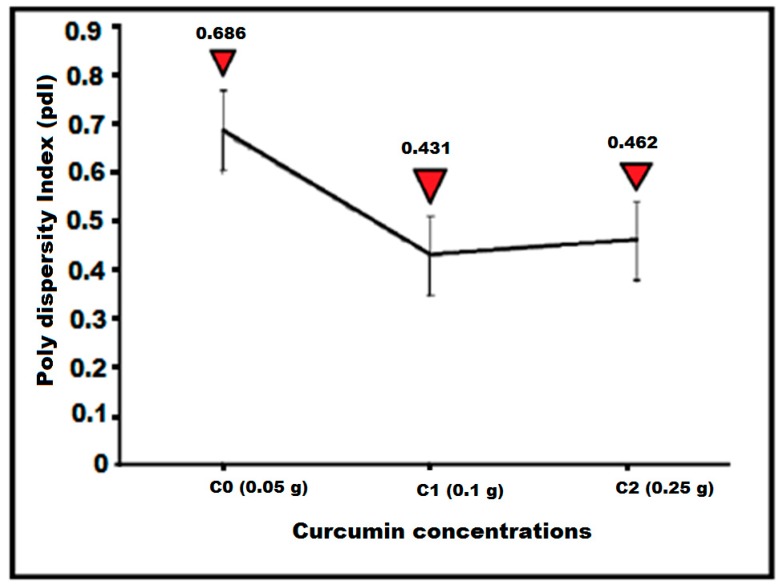
Polydispersity index (pdI) for various concentrations of curcumin, i.e., Curcumin, C0, C1, and C2.

**Figure 5 molecules-24-00719-f005:**
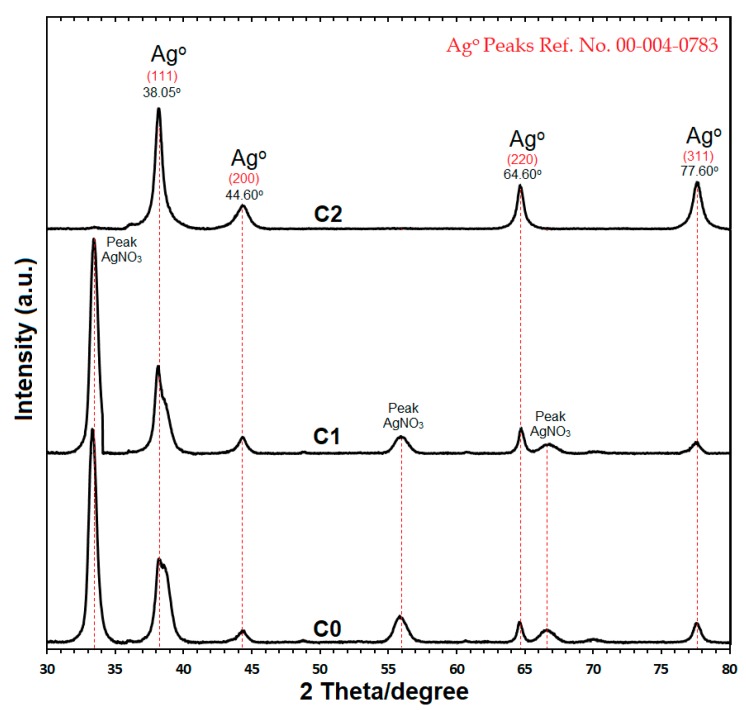
Powder X-ray diffraction pattern for C2, C1, and C0 samples.

**Figure 6 molecules-24-00719-f006:**
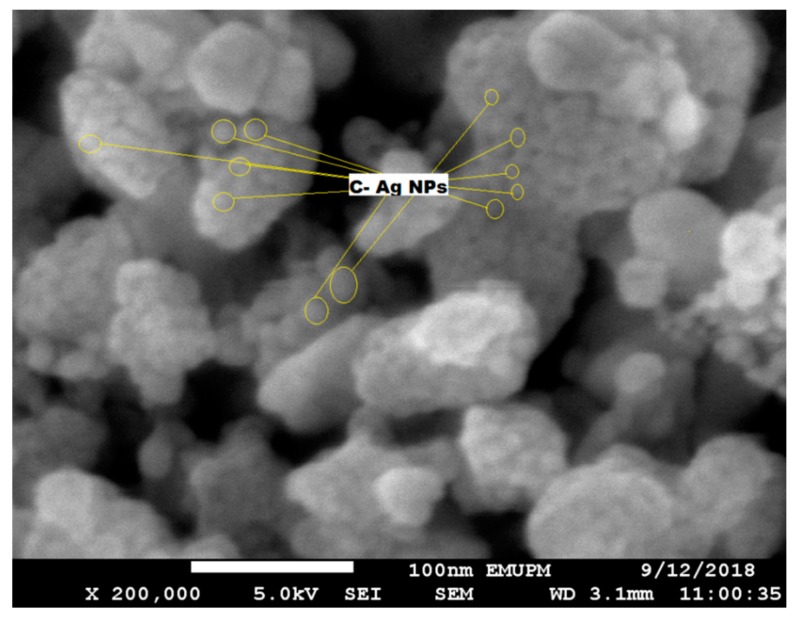
SEM micrographs of C-Ag NPs at 200,000× with prominent spherical NPs in the clusters.

**Figure 7 molecules-24-00719-f007:**
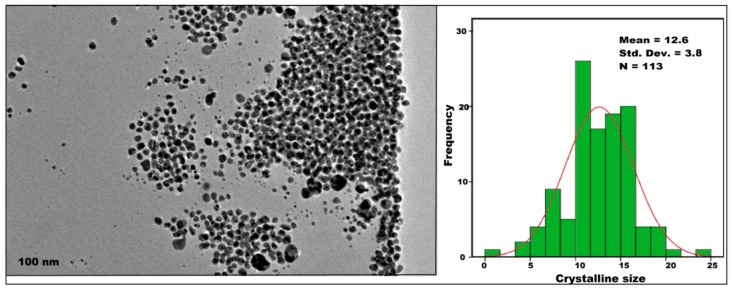
HRTEM analysis of C-Ag NPs from C2 (0.25 g) samples showing that NPs are spherical.

**Figure 8 molecules-24-00719-f008:**
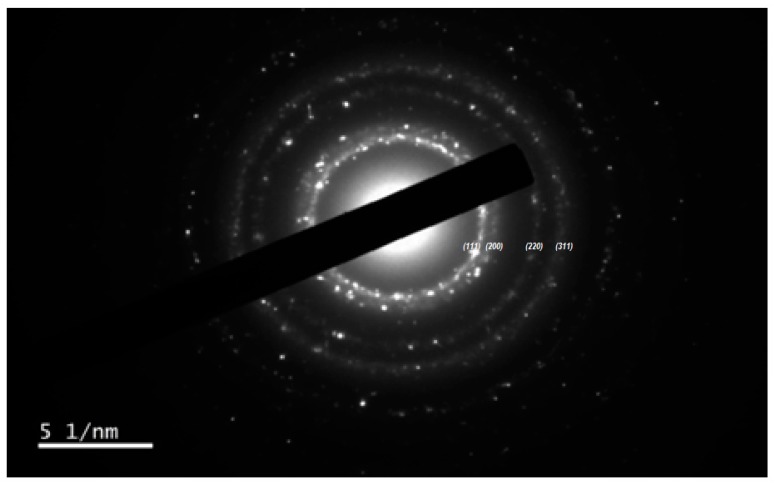
SAED pattern of C-Ag NPs from C2 (0.25 g) samples.

**Figure 9 molecules-24-00719-f009:**
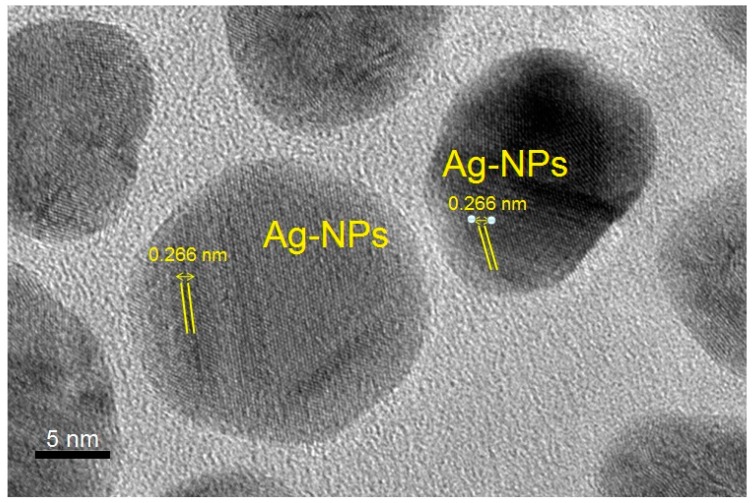
Lattice analysis of C-Ag NPs from C2 (0.25 g) samples. A thin layer of curcumin can be noticed around nanoparticles.

**Figure 10 molecules-24-00719-f010:**
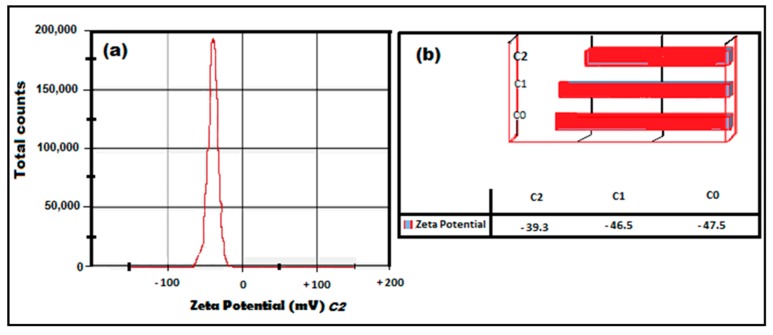
Zeta potential (mV) for curcumin concentrations (**a**) apparent zeta potential for C2 (**b**) zeta potential for C2, C1, and C0 concentrations.

**Table 1 molecules-24-00719-t001:** Characteristics of C-Ag NPs synthesized from the curcumin samples C2, C1, and C0.

No.	Sample Name	T	Poly Dispersity Index	ZP	Mob	Wave Length (nm)λ	pH
	C-Ag NPs	°C	(pdI)	mV	µmcm/Vs	-	-
**1**	C0	24.9	0.686	-	-	-	-
**2**	C0	25	-	−47.5	−3.7	427	9.66
**3**	C1	25	0.431	-	-	-	-
**4**	C1	25.1	-	−46.5	−3.6	428	9.78
**5**	C2	25.1	0.462	-	-	-	-
**6**	C2	25	-	−39.3	−3.0	445	9.92

pdI, zeta potential, mobility, wavelength and pH of C2, C1, and C0 samples.

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
