# Peer review of "Rapid Green Synthesis and Characterization of Silver Nanoparticles Arbitrated by Curcumin in an Alkaline Medium"

_molecules, 2019, doi:10.3390/molecules24040719_

Reviewer 1 Report

The article "Rapid Green Synthesis and Characterization of Silver Nanoparticles Arbitrated by Curcumin in the Alkaline medium" by Kumari et al. described the synthesis of AgNPs in alkaline medium using curcumin. The AgNPs were then extensively characterized using UV-Vis, polydispersity index, PXRD, FESEM, HRTEM, SAED and zeta potential. The role of the amount of curcumin used in the synthesis of AgNPs was also studied, which revealed that proper amount of curcumin was required to prepare good quality AgNPs. Studies on the preparation, characterization and application of AgNPs (in an environment-friendly manner) have been widely investigated and reported. Curcumin-assisted AgNPs synthesis were also reported, thus this article lacks of novelty and importance; further no bio-applications of the AgNPs were evaluated. Some comments are provided below:

1, the authors have used "silver nanoparticles" repeatedly in the paper, an abbreviation such as "AgNPs" might help;

2, lines 58 and 59, the authors claims " To our knowledge, the rapid green synthesis of silver nanoparticles in the alkaline medium, arbitrated by curcumin, has not been reported yet" in fact, a quick search with key words "curcumin silver nanoparticles" shown they have been various reports on this topic. These studies should be cited and the claim should be modified. 

3,lines 82-84, the sentence was not clear, please revise.

4, line 102, the authors states the AgNPs have good stability at room temperature "over the time", it would be better if a finite time frame could be provided during which AgNPs are stable.

5, lines 178-180, "All the colloidal solutions were recorded as stable as the Zeta potential (ZP) was recorded as– 47.5 mV (C0), â€“ 46.5 mV (C1) and â€“ 39.3 mV (C2) (Figure 10 b; Figure 10 a) which showed high stability of the nano suspension", the sentence is not clear, please revise.

Author Response

Reviewers

Reviewer’s Comments

Changes made?

Page Number

Remarks

Reviewer 1

Submission Date:

01 November 2018

Date of this review:

08 Nov 2018 08:02:22

1.       the authors have used "silver nanoparticles"   repeatedly in the paper, an abbreviation such as "AgNPs" might help

Yes

1, 2, 11

Revised as per   kind suggestion

2.       lines 58 and 59, the authors claims " To our   knowledge, the rapid green synthesis of silver nanoparticles in the alkaline   medium, arbitrated by curcumin, has not been reported yet" in   fact, a quick search with key words "curcumin silver nanoparticles"   shown they have been various reports on this topic. These studies should be   cited and the claim should be modified

Yes

2

Revised

3.       lines 82-84, the sentence was not clear, please revise

Yes

3

Revised

4.       line 102, the authors states the AgNPs have good stability at   room temperature "over the time", it would be better if a finite   time frame could be provided during which AgNPs are stable

Yes

3

Revised

5.       lines 178-180, "All the colloidal solutions were recorded   as stable as the Zeta potential (ZP) was recorded as– 47.5 mV   (C0), â€“ 46.5 mV (C1) and â€“ 39.3 mV (C2) (Figure 10 b;   Figure 10 a) which showed high stability of the nano suspension", the   sentence is not clear, please revise

Yes

9

Revised

Reviewer 2 Report

The work is very interesting in the field of food packaging, pharmaceutics and biomedicine.

However, control missing in some experiments. For this reasons, the paper requires a major revision. In particular the experimental evidence of curcumin in  the C-Ag-NPs is absent.

The reported thin layer of curcumin is not visible in the HRTEM micrographs of figures 7 and 9. I suspect that curcumin sublimated under the electron beam.

I suggest the use of infrared spectroscopy or the thermogravimetric analysis to evidenced and quantify curcumin in the C-Ag-NPs.

Some concerns with the manuscript are given below.

1) in row 26 (rows 133, 152 , 235-236,  fig 7 and table 1,) too many digit number are used. Accuracy and precision of the method used to measure the average size does not consent it. I suggest to indicate 12.6+-3.8 nm (or 13+-4 nm)

2) in rows 65 and 204 the molar ratio of NaOH is not correct (01M or 1 M ?);

3) rows 75-79:  does cited aldehyde group come from the chetoenolic group of the curcumin molecule?

4) in rows 82-84 and 205-208 the volume of all solution is not indicated

5) the quality of figure 5 is to be improved; The PXRD spectra of c2 sample is shifted to high angle (of about + 1° 2theta). A shift of 0.1 degree between experimental data and theoretical pdf card is reported: authors should be explain such discrepance.

6) The authors do not mention in the paragraph 3.3 the wavelength of the x radiation used (CuKalfa?). Is it correct to use the average value between alpha1 and alpha2 (1.5424 Angstrom) in the Scherrer equation?

7) Why does AgNO3 remains after washing?

9)The authors should clearly state that the FESEM and HRTEM measurements refer to the C2 sample

10)  The amount of C-Ag-NPs is not corrected (01 (?)  micrograms) in row 139

11) gm in fig 10b is uncorrected, indicate g otherwise indicate only C0/ C1/ C2

12) I suggest to delete the mean  values (last two rows of the table) in table 1 because it is inappropriate to average data of very different samples that contain different substances.

Author Response

Reviewers

Reviewer’s Comments

Changes made?

Page Number

Remarks

Reviewer 2

Submission Date:

01 November 2018

Date of this review:

19 Nov 2018 20:36:36

1.        in row 26 (rows 133, 152 , 235-236,  fig 7 and table 1,)   too many digit number are used. Accuracy and precision of the method used to   measure the average size does not consent it. I suggest to indicate 12.6+-3.8   nm (or 13+-4 nm)

Yes

1, 6, 7, 9,   11, 12

Revised as per   kind suggestion

2.        in rows 65 and 204 the molar ratio of NaOH is not correct (01M   or 1 M ?)

Yes

2, 11

The molar   ratio of NaOH is 1 M.

3.        rows 75-79:  does cited aldehyde group   come from the chetoenolic group of the curcumin molecule?

Yes

3

The aldehyde   group comes from methanolic structure of curcumin.

4.        in rows 82-84 and 205-208 the volume of all solution is not   indicated

Yes

3, 11

Revised as per   kind suggestion

5.      the quality of figure   5 is to be improved; The PXRD spectra of c2 sample is shifted to high   angle (of about + 1° 2theta). A shift of 0.1 degree between experimental data   and theoretical pdf card is reported: authors should be explain such   discrepance.

Yes

5

The figure is   replaced. The shift of 0.1 degree was produced when figure 05 was prepared.   It is now removed as per kind suggestion.

6.      The authors do not mention in the paragraph 3.3 the   wavelength of the x radiation used (CuKalfa?). Is it correct to use the   average value between alpha1 and alpha2 (1.5424 Angstrom) in the Scherrer   equation?

Yes

6

1.5424 is the X-Ray value of PXRD equipment used to   analyze the powder sample.This constant can be used in  Debye- Scherrer equation to estimate   crystalline size of Ag NPs.

7.        Why does AgNO3 remains after washing?

Yes

5

The amount of   curcumin oxide (0.75 ml) might not be quite enough in C0 and C1 samples for   the reduction of AgNO3 into AgË– ions. Due to which AgNO3 peaks   appeared in PXRD results.

8.        The authors should clearly state that the FESEM and HRTEM   measurements refer to the C2 sample

Yes

7

Revised as per   kind suggestion

9.        The amount of C-Ag-NPs is not corrected (01 (?)    micrograms) in row 139

Yes

6

Revised as per   kind suggestion

10.     gm in fig 10b is uncorrected, indicate g otherwise indicate only   C0/ C1/ C2

Yes

9

Revised as per   kind suggestion

11.     I suggest tobdelete the mean values (last two rows of the table)   in table 1 because it is inappropriate to average data of very different   samples that contain different substances.

Yes

11

Revised as per   kind suggestion

Reviewer 3 Report

Dear Authors,

I have several comments:

- Ag is toxic at certain concentrations, and you should comment this aspect. 

- You presented a good characterization of the nanomaterial, but without detailing its biological activity. You should introduce the presentation of biological activities before claiming  that AgNPs may be used in the food industry, for example. 

- You should detail the possible implication in the synthesis of curcumin isomers and the influence of the alkaline environment in the reduction process. 

- The level of the majority compound is important and, therefore, a nanomaterial yield should be considered. 

- You should present whether  the silver or the curcumin determines the biological effect,and how it is expressed.

 Best regards!

Author Response

Reviewers

Reviewer’s Comments

Changes made?

Page Number

Remarks

Reviewer 3

Submission Date:

01 November 2018

Date of this review:

09 Nov 2018 19:36:40

1.        Ag is toxic at certain concentrations, and you should comment   this aspect

Yes

2

Commented in Introduction   as per kind suggestion

2.        You presented a good characterization of the nanomaterial, but   without detailing its biological activity. You should introduce the   presentation of biological activities before claiming  that AgNPs may be   used in the food industry, for example. 

Yes

2

Presented in   the introduction

3.        You should detail the possible implication in the synthesis of   curcumin isomers and the influence of the alkaline environment in the   reduction process. 

Yes

3

Row 100-101.

4.        The level of the majority compound is important and, therefore,   a nanomaterial yield should be considered. 

Yes

3

Row 104-105.

5.        You should present whether the silver or the   curcumin determines the biological effect, and how it is expressed.

Yes

2

Row 48-49.

Round  2

Reviewer 1 Report

The authors have addressed some of the concerns presented in the previous version of the manuscript, however again, studies on the preparation, characterization and application of AgNPs (in an environment-friendly manner) have been widely investigated and reported. Curcumin-assisted AgNPs synthesis were also reported (the authors did not cite the relevant references), thus this article lacks of novelty and importance; further no applications of the AgNPs were demonstrated. 

Author Response

Honorable reviewer,

Good evening.

First of all we appreciate your kind and valuable suggestions to improve the quality of the manuscript. It provided us a chance to learn a lot. 

As for as the English language concern, the manuscript has been gone through an overall language and spell check by the experts and is ready for further processing.

The previous studies regarding the preparation, characterization and application of Ag NPs (in an environment-friendly manner) have also been incorporated ( references No. 20, 21, 22, 23, 24, 25, 26, 27, 33, and 36 may be considered). 

Our study is based on the synthesis and characterization of Ag NPs in an alkaline medium with safe, green, and relatively smaller Ag NPs mediated by curcumin, as curcumin is more sensitive to pH. 

Further the biological application of C -Ag NPs is planned for food preservation due to their smaller size and will be reported soon.

Regards and thanking in anticipation,

Authors. 

Reviewer 2 Report

The paper is now acceptable as it is

Author Response

Respected Reviewer,

I am grateful for your kind cooperation and valuable suggestions to improve the manuscript.

Kind regards.

Reviewer 3 Report

Dear authors,

After the second review I do not have any supplementary suggestions. 

Best regards!

Author Response

Honorable Reviewer,

As an author, I really appreciate your kind and valuable suggestions to improve the manuscript.

Kind regards.